# Dysregulation of Muscle-Specific MicroRNAs as Common Pathogenic Feature Associated with Muscle Atrophy in ALS, SMA and SBMA: Evidence from Animal Models and Human Patients

**DOI:** 10.3390/ijms22115673

**Published:** 2021-05-26

**Authors:** Claudia Malacarne, Mariarita Galbiati, Eleonora Giagnorio, Paola Cavalcante, Franco Salerno, Francesca Andreetta, Cinza Cagnoli, Michela Taiana, Monica Nizzardo, Stefania Corti, Viviana Pensato, Anna Venerando, Cinzia Gellera, Silvia Fenu, Davide Pareyson, Riccardo Masson, Lorenzo Maggi, Eleonora Dalla Bella, Giuseppe Lauria, Renato Mantegazza, Pia Bernasconi, Angelo Poletti, Silvia Bonanno, Stefania Marcuzzo

**Affiliations:** 1Neurology IV–Neuroimmunology and Neuromuscular Diseases Unit, Fondazione IRCCS Istituto Neurologico Carlo Besta, Via Celoria 11, 20133 Milan, Italy; claudia.malacarne@istituto-besta.it (C.M.); eleonora.giagnorio@istituto-besta.it (E.G.); paola.cavalcante@istituto-besta.it (P.C.); franco.salerno@istituto-besta.it (F.S.); francesca.andreetta@istituto-besta.it (F.A.); lorenzo.maggi@istituto-besta.it (L.M.); renato.mantegazza@istituto-besta.it (R.M.); pia.bernasconi@istituto-besta.it (P.B.); silvia.bonanno@istituto-besta.it (S.B.); 2PhD Program in Neuroscience, University of Milano-Bicocca, Via Cadore 48, 20900 Monza, Italy; 3Dipartimento di Scienze Farmacologiche e Biomolecolari, Centro di Eccellenza sulle Malattie Neurodegenerative, Università degli Studi di Milano, Via Balzaretti, 9, 20133 Milano, Italy; rita.galbiati@unimi.it; 4Molecular Neuroanatomy and Pathogenesis Unit, Neurology VII—Clinical and Experimental Epileptology Unit, Fondazione IRCCS Istituto Neurologico Carlo Besta, Via Celoria 11, 20133 Milan, Italy; cinzia.cagnoli@istituto-besta.it; 5Dino Ferrari Centre, Neuroscience Section, Department of Pathophysiology and Transplantation (DEPT), University of Milan, Via Francesco Sforza 35, 20122 Milan, Italy; mm.taiana@gmail.com (M.T.); stefania.corti@unimi.it (S.C.); 6Neurology Unit, IRCCS Foundation Ca’ Granda Ospedale Maggiore Policlinico, Via Francesco Sforza 35, 20122 Milan, Italy; monica.nizzardo1@gmail.com; 7Unit of Medical Genetics and Neurogenetics, Fondazione IRCCS Istituto Neurologico Carlo Besta, Via Celoria 11, 20133 Milan, Italy; viviana.pensato@istituto-besta.it (V.P.); anna.venerando@istituto-besta.it (A.V.); cinzia.gellera@istituto-besta.it (C.G.); 8Unit of Rare Neurodegenerative and Neurometabolic Diseases, Department of Clinical Neurosciences, Fondazione IRCCS Istituto Neurologico Carlo Besta, Via Celoria 11, 20133 Milan, Italy; silvia.fenu@istituto-besta.it (S.F.); davide.pareyson@istituto-besta.it (D.P.); 9Developmental Neurology Unit, Fondazione IRCCS Istituto Neurologico Carlo Besta, Via Celoria 11, 20133 Milan, Italy; riccardo.masson@istituto-besta.it; 10Neuroalgology Unit, Fondazione IRCCS Istituto Neurologico Carlo Besta, Via Celoria 11, 20133 Milan, Italy; eleonora.dallabella@istituto-besta.it (E.D.B.); giuseppe.lauriapinter@istituto-besta.it (G.L.); 11Department of Biomedical and Clinical Sciences “Luigi Sacco”, University of Milan, Via G.B. Grassi 74, 20157 Milan, Italy

**Keywords:** motor neuron diseases, muscle-specific microRNAs, amyotrophic lateral sclerosis, spinal muscular atrophy, spinal bulbar muscular atrophy, mouse models

## Abstract

Motor neuron diseases (MNDs) are neurodegenerative disorders characterized by upper and/or lower MN loss. MNDs include amyotrophic lateral sclerosis (ALS), spinal muscular atrophy (SMA), and spinal and bulbar muscular atrophy (SBMA). Despite variability in onset, progression, and genetics, they share a common skeletal muscle involvement, suggesting that it could be a primary site for MND pathogenesis. Due to the key role of muscle-specific microRNAs (myomiRs) in skeletal muscle development, by real-time PCR we investigated the expression of miR-206, miR-133a, miR-133b, and miR-1, and their target genes, in G93A-SOD1 ALS, Δ7SMA, and KI-SBMA mouse muscle during disease progression. Further, we analyzed their expression in serum of *SOD1*-mutated ALS, SMA, and SBMA patients, to demonstrate myomiR role as noninvasive biomarkers. Our data showed a dysregulation of myomiRs and their targets, in ALS, SMA, and SBMA mice, revealing a common pathogenic feature associated with muscle impairment. A similar myomiR signature was observed in patients’ sera. In particular, an up-regulation of miR-206 was identified in both mouse muscle and serum of human patients. Our overall findings highlight the role of myomiRs as promising biomarkers in ALS, SMA, and SBMA. Further investigations are needed to explore the potential of myomiRs as therapeutic targets for MND treatment.

## 1. Introduction

Motor neuron diseases (MNDs) are a heterogeneous group of rare disorders characterized by degeneration of motor neurons. The spectrum encompasses different phenotypes, depending on the involvement of upper and/or lower motor neurons [1,2]. MNDs include amyotrophic lateral sclerosis (ALS), spinal muscular atrophy (SMA), and spinal and bulbar muscular atrophy (SBMA).

ALS is a progressive and fatal neurodegenerative disease affecting both upper and lower motor neurons of the motor cortex, brainstem, and spinal cord. Motor neuron degeneration leads to muscle atrophy and death in approximately 3–5 years from onset, mainly due to respiratory failure [3]. Most ALS cases are sporadic (sALS), but 5–10% present familial inheritance (fALS), and 20% of these are due to mutations in the superoxide dismutase 1 (*SOD1*) gene, which acquires a toxic gain of function [4]. In both sALS and fALS, there is a variable loss of upper and lower motor neurons [5], ultimately resulting in a similar pathology [6].

Spinal muscular atrophy (SMA) is an autosomal recessive neuromuscular disorder characterized by selective loss of the brainstem and spinal motor neurons, leading to progressive amyotrophic paralysis, respiratory deficiency, and death in more severe cases. The disease is due to mutations in survival motor neuron (*SMN)* 1 gene, resulting in a truncated SMN protein, but deletion mutations can also result in a complete lack of SMN protein. The paralogous *SMN2* gene partially compensates full-length SMN protein production, mitigating the phenotype. Severity and onset of the disease are inversely related to the *SMN2* copy numbers [7]. Notably, the ubiquitously expressed SMN protein acts as an assembly factor for small nuclear ribonucleoproteins (snRNPs) involved in mRNA splicing and pre-mRNA maturation [8].

Although ALS and SMA present differences in onset and genetic causes, it is reported that the two diseases have a strong interlink since FUS and mutant SOD1 directly interact with SMN complex. *Smn* depletion aggravates disease progression in ALS mice, and duplications of *SMN1* were linked to sALS. Beyond genetic interactions, growing evidence further suggests that both diseases share common pathological identities such as intrinsic muscle defects [9]. It is crucial to optimize developing viable ALS gene therapies considering and referring those already approved for SMA [10].

SBMA is an inherited X-linked MND characterized by lower motor neuron loss. It is caused by an expansion of the CAG repeat sequence in the exon 1 of the androgen receptor (*AR*) gene coding for an elongated polyglutamine (polyQ) tract in the N-terminus of the AR protein (ARpolyQ) [11]. The *AR* with the CAG expansion gains a neurotoxic function probably causative of the disease [12]. The aberrant polyQ-protein misfolds and leads to the formation of aggregates that alter proteosomal and authophagic activities in affected cells, and are known to induce neuronal dysfunctions [13,14].

Despite differences in onset, progression, and genetic causes, several data revealed a common involvement of skeletal muscle in MNDs. Whether muscle tissue can be a primary site for disease pathogenesis is still largely debated and not completely clarified. However, growing evidence strongly suggests that muscle degeneration in MNDs is not only due to motor neuron death, but also to intrinsic changes in myocytes, and to alterations in muscle structure and function that may be responsible for variations in disease onset and/or progression [14,15,16,17,18]. Indeed, while it is generally accepted that ALS is caused by motor neuron death, several studies in humans and mouse models showed neuromuscular junction and muscle dysfunction at disease stages in which motor neuron loss was not yet detected [16,19,20,21,22], suggesting that this could represent a key step in ALS pathogenesis [21,23,24]. Moreover, MRI studies demonstrated that muscle wasting is one of the earliest events detectable in the G93A-SOD1 animal model [16,17], supporting the idea that skeletal muscle might be an additional trigger of ALS disease, and not only a passive player [16,23,25].

Interestingly, several studies suggested that RNA processing defects related to SMN deficiency could lead to breakdown of the neuromuscular system [26] or alterations in muscle-specific functions [27,28]. Muscle from severe SMA patients shows widespread small myofibers, which have a developmental arrested appearance [29] and an immature expression profile of myofibrillary proteins [30]. The most common SMA animal model is the Δ7SMA mice lacking exon 7 in the *Smn* gene, expressed under control of the human *SMN2* promoter [31]. Experiments in SMA mice indicated that *Smn* deficiency causes intrinsic defects in muscle development by impairing myofiber growth and regeneration, thus contributing to SMA phenotype [32,33,34], even preceding spinal motor neuron damage [35].

Evidence suggests that muscle cells are primarily affected also in SBMA and are crucial components of the disease pathogenesis [36,37,38,39,40]. Indeed, silencing the peripheral mutant ARpolyQ expression in different SBMA mouse models resulted in prolonged survival, thereby providing evidence for a direct effect of ARpolyQ on muscle atrophy [12,41].

Data indicate that alteration of RNA metabolism is a crucial event in MNDs, and microRNAs (miRNAs) seem to be highly implicated [42,43], since almost all aspects of skeletal muscle development are regulated by miRNAs [44,45]. Myogenic miRNAs, also known as myomiRs, display a muscle enriched expression pattern and are critical modulators of the myogenic program by targeting a wide range of muscle genes [46,47]. MyomiRs, including miR-206, miR-133a, miR-133b, and miR-1, act on the mRNAs of genes acting as potent promoters of muscle proliferation and regeneration, such as targeting paired box 7 (*PAX7*) gene, and they foster myogenic differentiation by regulating myogenin (*MYOG*), myoblast determination protein 1 (*MYOD1*), and myocyte enhancer factor-2 (*MEF2*) genes [44,46,48,49,50,51]. Thus, their broad involvement in several pathophysiological conditions, including muscle degeneration or regeneration in MNDs [18,52,53,54], is not surprising. For example, overexpression of miR-206 promotes muscle regeneration during ALS progression and contributes to the maintenance and regeneration of neuromuscular synapses via regulation of neuromuscular gene expression [52,55]. In G93A-SOD1 mice and ALS patients, miR-206 is up-regulated during disease progression, likely as a compensatory mechanism, albeit not sufficient to counteract neurodegeneration [56,57]. MiR-206 is also increased in SMA mouse muscle tissue, possibly with a protective role in response to the severe impairment of neuromuscular junction [18]. MiR-133a and miR-133b are useful molecular markers of muscle differentiation and atrophy [54,58]. Indeed, they are involved in the regulation of several processes during skeletal muscle development as observed both in ALS mouse model and patients [52,59,60,61]. However, their precise molecular mechanisms accountable for muscle alterations remain mostly unknown, and the involvement of these molecules in the progression of MNDs, particularly in SMA and SBMA, deserves to be thoroughly investigated. MiR-1 is implicated in myogenic differentiation mainly by acting on *Pax7* [44,48]. Recently, a reduction in miR-1 expression in muscle biopsies of ALS patients was described, along with miR-133a, suggesting a broad dysfunction both in the myogenic and differentiation processes [60]. Based on this knowledge, more extensive studies on animal models and humans are needed to better understand the involvement of myomiRs in muscle remodeling in ALS, SMA, and SBMA during disease progression, and their potential as molecular targets of effective treatments for these conditions.

In this study, we analyzed the expression of miR-206, miR-133a, miR-133b, and miR-1 in ALS, SMA, and SBMA mouse models, considering different disease stages. Moreover, to corroborate the potential role of these molecules as noninvasive biomarkers, we analyzed their expression in serum samples of *SOD1*-mutated ALS, SMA, and SBMA patients. Our findings show a dysregulation of myomiR expression in muscle of ALS, SMA, and SBMA animals, revealing a common specific pathogenic mechanism associated with muscle atrophy in these diseases. Similar alterations in myomiR expression were observed in serum samples of human patients, suggesting an implication of these molecules in MNDs and their potential as promising noninvasive biomarkers and/or future therapeutic targets to modulate disease course.

## 2. Results

### 2.1. Dysregulated Expression of MyomiRs in G93A-SOD1, Δ7SMA, and AR113Q Mouse Muscle Tissue in Relation to Disease Progression

To verify whether myomiR dysregulation in skeletal muscle tissue may be a common pathogenic mechanism in ALS, SMA, and SBMA, we assessed the expression levels of miR-206, miR-133a, miR-133b, and miR-1 in G93A-SOD1, Δ7SMA, and AR113Q mice at presymptomatic, onset, and symptomatic phases of disease.

MiR-206 levels were found to be significantly increased in symptomatic phase of disease in affected mice of all the three animal models compared to that of control mice. G93A-SOD1 mice displayed an increase of miR-206 starting from week 8, corresponding to the presymptomatic phase, to week 18, suggesting that its overexpression is an early phenomenon appearing before disease symptoms become evident, and maintained during disease course (as illustrated in Figure 1A, * *p* < 0.05); in Δ7SMA and AR113Q mice, miR-206 up-regulation was significant at symptomatic disease phase, corresponding to day 10 and week 40, respectively (as illustrated in Figure 1B,C, * *p* < 0.05 for Δ7SMA, and ** *p* < 0.01 for AR113Q mice vs. controls). In contrast, miR-133a and miR-133b levels in G93A-SOD1 and AR113Q mice decreased during symptomatic disease phase compared to that of control animals (as illustrated in Figure 1A,C). Indeed, miR-133a was down-regulated from week 15 to 18, and miR-133b from week 12 (disease onset) to 18 in G93A-SOD1 (as illustrated in Figure 1A, * *p* < 0.05, and ** *p* < 0.01), and both miRNAs were significantly decreased at week 40 in AR113Q mice (as illustrated in Figure 1C, * *p* < 0.05 for miR-133a, and ** *p* < 0.01 for miR-133b). In Δ7SMA mice, miR-133a and miR-133b showed an opposite trend, since both the two miRNAs were up-regulated at symptomatic disease phase (day 10), although differences did not reach statistical significance for miR-133b (as illustrated in Figure 1B; ** *p* < 0.01 for miR-133a). MiR-1 expression showed a trend similar to that of miR-133a and miR-133b: it was significantly decreased in G93A-SOD1 and AR113Q affected mice, but increased in Δ7SMA mice. Specifically, lower levels of this miRNA were found in G93A-SOD1 animals from week 8 to 18, and in AR113Q mice from week 14 to 40 (as illustrated in Figure 1A,C, * *p* < 0.05, and ** *p* < 0.01), whereas a significant miR-1 increase was observed at day 10 in Δ7SMA mice (** *p* < 0.01). In line with data observed in the three MND animal models, we discovered significantly increased levels of miR-206 in denervated mice (as illustrated in Appendix A; * *p* < 0.05), whereas miR-133a, miR-133b, and miR-1 levels were decreased, as observed in G93A-SOD1 and AR113Q animals (as illustrated in Appendix A; * *p* < 0.05). Our findings suggest that the dysregulated expression of myomiRs could be significantly associated with the response to muscle wasting which characterizes MNDs.

### 2.2. Altered Expression of MyomiR Targets in G93A-SOD1, Δ7SMA, and AR113Q Muscle Tissue in Relation to Disease Progression

Based on the miRWalk 2.0 and miRBase database of predicted and validated miRNA-target interactions [62] and on literature data reporting experimentally validated myomiR targets [44,46,48,49,50,51], we selected the following myomiR target genes for analysis in MND muscle tissue: *Pax7*, known to be responsible of muscle cell proliferation, *Myog*, *Myod1*, and *Mef2a*, known to be implicated in muscle cell differentiation. These four genes are targets of all the myomiRs considered in this study. Their expression levels were quantified in G93A-SOD1, Δ7SMA, and AR113Q mouse muscles throughout disease progression, as previously performed for myomiR analysis. In G93A-SOD1 mice, *Pax7* mRNA levels were not significantly different from those of control mice at the different disease stages, whereas *Myod1* and *Myog* expression was significantly up-regulated from week 12 to 18 (as illustrated in Figure 2A, * *p* < 0.05 for *Myod1* at week 12 and 15, and ** *p* < 0.01 for *Myod1* at week 18 and *Myog* at week 12, 15, and 18). On the contrary, *Mef2a* transcriptional levels were significantly reduced in affected mice at symptomatic disease stages (15 and 18 weeks) compared to controls (as illustrated in Figure 2A, ** *p* < 0.01 for week 15 and * *p* < 0.05 for week 18). Δ7SMA animals showed comparable expression levels of *Pax7*, *Myod1* and *Mef2a* to those of controls, but they were characterized by a significant overexpression of *Myog* at day 2 (as illustrated in Figure 2B, ** *p* < 0.01). Increased expression levels of *Pax7* were found in AR113Q compared to that of control mice at both week 14 and 40 (as illustrated in Figure 2C, * *p* < 0.05); *Myod1* and *Myog* mRNA levels were increased at week 8, with *Myog* expression being also increased at week 40 when *Myod1* appeared down-regulated (as illustrated in Figure 2C, * *p* < 0.05). Lastly, *Mef2a* mRNA levels significantly decreased in AR113Q mouse muscle at symptomatic disease stage (40 weeks) compared to controls (as illustrated in Figure 2C, * *p* < 0.05). By western blot, we analyzed the expression of PAX7 and MEF2A protein levels in the muscle tissue of the three animal models and relative controls along with MYOG, which was chosen since it functions downstream and is synergic with MYOD1 at the terminal stages of myogenic differentiation [63,64].

In line with gene expression data, no significant results were obtained for PAX7 expression in G93A-SOD1 compared to that of control mice, although the protein showed an increasing trend with disease progression (as illustrated in Figure 3A). Moreover, MYOG protein levels augmented starting from week 12 to 18 (as illustrated in Figure 3A, * *p* < 0.05) in accordance with transcriptional data. A decreasing trend of MEF2A protein level was detected in G93A-SOD1 mice compared to that of controls, but it did not reach statistical significance. In the Δ7SMA model, we detected a significant decrease of PAX7 protein in the affected compared to that of control mice at day 10, accompanied by an increase of MEF2A protein (as illustrated in Figure 3B, * *p* < 0.05). In line with the increased *Pax7* mRNA levels previously observed at week 14 and 40 (as illustrated in Figure 2C), we revealed an increase of the PAX7 protein in the AR113Q compared to that of control mice at week 40 (as illustrated in Figure 3C, * *p* < 0.05). At the same week, the AR113Q-affected mice presented a significant increase of MYOG protein (as illustrated in Figure 3C, * *p* < 0.05), in accordance with the increased gene transcriptional levels previously shown (as illustrated in Figure 2C). MEF2A protein levels did not change during disease progression in AR113Q mice (as illustrated in Figure 3C). A slight reduction of *Pax7* mRNA levels was found in denervated mouse model compared to that of controls, accompanied by increased transcriptional levels of *Myod1*, *Myog*, and *Mef2a*, although differences were not confirmed at protein level, with increased PAX7 and unchanged levels of MYOG and MEF2A (as illustrated in Appendix A). However, since the number of samples included in the western-blot analyses was limited, the protein data may be potentially underpowered.

### 2.3. MyomiR-mRNA Target Correlations in Muscle Tissue of G93A-SOD1, Δ7SMA, and AR113Q Mice

In Figure 4 and Appendix A, we showed the relationship between the expression of myomiRs and their putative target genes, estimated as mean of quantitative real-time PCR values, obtained at the different disease stages in the muscle tissue of MNDs and denervated animals compared to that of the relative controls. Figure 4 and Appendix A illustrate possible correlations between up-and down-regulated myomiRs and the up-and down-regulated target genes in each animal model considered. Spearman’s correlation analyses were then performed to correlate the expression levels of each myomiR and the mRNA levels of each putative target gene in muscle tissue of G93A-SOD1, Δ7SMA, and AR113Q mice, considering all the time points (as illustrated in Appendix A). Since miRNAs act as negative and positive regulators [44,46,48,49,50,51], we focused on both inverse and direct correlations, considering coefficient r lower than −0.5 or higher than +0.5 as good inverse or direct correlation, respectively. The most conspicuous, significant correlations were found for the following myomiR/mRNA pairs: miR-206/*Pax7* and miR-206/*Myod1* in G93A-SOD1 mice (as illustrated in Figure 5A); miR-133a/*Pax7*, miR-1/*Pax7*, and miR-1/*Mef2a* in Δ7SMA mice (as illustrated in Figure 5B); miR-133a/*Myod1*, miR-133b/*Myod1*, miR-133b/*Myog*, miR-1/*Pax7,* miR-1/*Myod1*, and miR-1/*Myog* in AR113Q mice (as illustrated in Figure 5C). In denervated mice we did not observe significant negative or positive correlations between each myomiR and the putative targets (*p* > 0.05).

### 2.4. Alteration of G93A-SOD1, Δ7SMA, and AR113Q Muscle Architecture at Late Disease Stage

To evaluate muscle atrophy, we performed histological analysis of gastrocnemius muscle specimens in the three MND animal models at late disease stage when the animals manifested the most significant molecular changes. Morphological analysis by hematoxylin/eosin staining showed that skeletal muscles from G93A-SOD1 (as illustrated in Figure 6A), Δ7SMA (as illustrated in Figure 6B), and AR113Q mice (as illustrated in Figure 6C) were characterized by the presence of smaller fibers, larger connective endomysial spaces, and few fibers with centrally located nuclei, compared to that of those present in age-matched controls. As we already showed in G93A-SOD1 animal model [16] and in agreement with a previous report [14], a trend in increased muscle disorganization was evident at late disease stage. In G93A-SOD1, we confirmed a significant reduction of muscle fiber diameters compared to that of control (as illustrated in Figure 6A, * *p* < 0.05). A similar trend of reduction of muscle fiber diameter was observed in Δ7SMA mice compared to that of control at day 10 (as illustrated in Figure 6B); in this mouse model muscle fiber diameter was generally smaller compared to that of the other MND animal models due to the lower age at the time of analysis. Similar results were observed in AR113Q mice compared to control at week 40 (as illustrated in Figure 6C). In line with data observed in the three MND animal models, we detected similar morphological changes also in the denervation mouse model (as illustrated in Appendix A).

### 2.5. Altered MyomiR Expression in Serum Samples of ALS, SMA, and SBMA Patients

We assessed the expression levels of miR-206, miR-133a, miR-133b and miR-1 in serum of mutant-*SOD1*-ALS, pediatric SMA, and SBMA patients (as illustrated in Table 1), to evaluate the potential of these myomiRs as noninvasive clinical biomarkers in the human pathologies. We found a significant increase of miR-206 levels in serum of ALS and SMA patients, while in SBMA sera miR-206 expression showed the same increasing trend, but the difference between patients and controls was not significant (as illustrated in Figure 7A). By ROC curve analysis we obtained sensitivity and specificity diagnostic performance results which support a possible role for circulating miR-206 as disease biomarker for ALS and SMA (as illustrated in Figure 7B). Regarding miR-133a, miR-133b, and miR-1 detection in serum, we did not find significant differences in ALS, SMA, and SBMA patients compared to that of healthy controls (as illustrated in Appendix A).

## 3. Discussion

Recently, it was suggested that muscle degeneration in MNDs is not only due to motor neuron death since intrinsic changes in muscle cells may be directly responsible for disease onset and/or progression [14,15,16,18,65]. Notably, almost all aspects of skeletal muscle development are regulated by miRNAs, and specifically by myomiRs that are selectively enriched in muscle tissue [44,48,66]. Here, we analyzed the expression of several myomiRs crucial for muscle function (miR-206, miR-133a, miR-133b, and miR-1) along with their putative target genes in muscle tissue of ALS, SMA, and SBMA animal models during disease progression. Next, we extended these observations to the serum of human patients. These approaches were taken to: (i) understand whether common or specific molecular mechanisms involving these myomiRs may underlie muscle impairment in the foremost MNDs, and (ii) demonstrate the potential role of myomiRs as noninvasive clinical biomarkers for MNDs.

Our data confirmed that miR-206 levels are increased in skeletal muscle of ALS and SMA animal models compared to that of controls at the symptomatic phase of disease, and proved that this is a feature also shared by SBMA mice. In G93A-SOD1 mice, this increment was already significant at week 8 (presymptomatic disease phase) and was maintained until week 18 (symptomatic phase) as previously observed [52,67,68], suggesting a protective role for miR-206 by inducing muscle regeneration during ALS progression to counteract muscle wasting. MiRNAs act as negative modulators of mRNA, degrading the target genes by miRNA-RISC complex [69], or as positive modulators, indicating the existence of feed forward regulation mediated by transcription factors [70]. We identified a positive correlation between miR-206 levels and the transcription levels of *Myod1*, a gene encoding a transcription factor crucial for muscle myogenesis [50] in G93A-SOD1 mouse muscle. A negative correlation was found between the miR-206 levels and *Pax7*, a gene implicated in satellite cell proliferation [48]. This supports the potential role of miR-206 in promoting MYOD1-driven myogenesis while inhibiting muscle stem cell proliferation via PAX7. Further, MYOD1 itself may induce miR-206 expression, establishing a positive regulatory feed forward loop which potentiates MYOD1-mediated muscle cell differentiation [50,60]. We also showed a significant increase of MYOG expression at both transcriptional and protein level at symptomatic disease phase in G93A-SOD1 muscle tissue, suggesting that this molecule could contribute to the regenerative attempt by regulating miR-206 expression [71,72]. In agreement with previous works [18,73], an up-regulation of miR-206 was also found in Δ7SMA mice compared to that of controls. Moreover, we analyzed for the first time the expression levels of myomiRs in the SBMA animal model, also demonstrating an up-regulation of miR-206 at the symptomatic disease stage, compared to that of controls. These findings unequivocally disclose that miR-206 up-regulation is a common pathogenic feature associated with muscle atrophy occurring throughout MND course, although further investigation is needed to depict functional miR-206/target gene interactions in relation to specific disease pathogenesis and progression.

As regards to the other investigated myomiRs, we revealed a significant reduction of miR-133a levels in G93A-SOD1 and in AR113Q mice at symptomatic phase of disease. Notably, miR-133a levels are also reduced in skeletal muscle biopsies obtained from ALS subjects at baseline and after 12 weeks of disease progression [60]. In the fine balance between myocyte proliferation and differentiation, miR-133 mainly supports muscle stem cell proliferation [45]. Thus, miR-133 reduction, synergistically with miR-206 increase, might be in line with the regenerative attempt towards myoblast differentiation instead of their proliferation. Further, in AR113Q mice, *Myod1* mRNA levels were reduced at the symptomatic disease phase, and a significant positive correlation between miR-133a and *Myod1* was found, pointing at a possible relationship between miR-133a down-regulation and reduced *Myod1* expression. Indeed, dysregulation of miR-133a and *Myod1* in AR113Q mice may inhibit the myogenic differentiation processes [45], causing a pathogenic event that prevents a regenerative compensatory attempt by miR-206. In skeletal muscle, MYOD1 acts synergistically with MYOG and MEF2A in myomiR regulation to preserve muscle homeostasis [71]. This is strengthened by our molecular data showing that the expression of *Mef2a*, another gene physiologically involved in promoting muscle cell differentiation [45,51], is reduced in AR113Q mice. Both MYOD1 and MEF2A positively regulate the expression of the miR-1/miR-133 cluster [45], and *Myod1* and *Mef2a* reduction could contribute to the down-regulation of miR-133a observed in AR113Q mice. By contrast, we found an up-regulation of miR-133a associated with muscle atrophy in Δ7SMA mice compared to that of controls. Based on our myomiR/mRNA correlation analysis, we found a significant negative correlation between miR-133a and *Pax7* mRNA levels in Δ7SMA mice. MiR-133a enhancement may reduce *Pax7* expression, and therefore cell proliferation, limiting the tissue ability to sustain and improve muscle condition. This hypothesis is supported by our biochemical data showing a decrease of PAX7 protein levels in Δ7SMA mice compared to that of controls, and it is in line with the concept that SMA is more a muscle development disorder compared to that of the other MNDs [29,35].

As well as miR-133a, miR-133b levels were also reduced in muscle tissue of G93A-SOD1 and AR113Q mice, but not in Δ7SMA animal model compared to controls at symptomatic disease phase. This agrees with a previous study showing an opposite trend of expression of miR-133b and miR-206 [59], although the two miRNAs are on the same bicistronic cluster [46]. *Myod1* and *Mef2a* mRNA levels were decreased in AR113Q animals, whereas *Myog* resulted up-regulated, and a significant positive correlation was found between miR-133b levels and both *Myod1* and *Myog* mRNA in these mice, supporting once again the idea that the regenerative attempt of miR-206 is not sustained by miR-133b.

MiR-1 constitutes a bicistronic cluster with miR-133a [44], and it is also down-regulated in G93A-SOD1 and AR113Q and up-regulated in Δ7SMA mice at symptomatic disease stage, suggesting that decreased expression levels of the two myomiRs may inhibit myogenic regenerative processes. A similar reduction of miR-1 expression was observed by Jensen and colleagues (2016) in ALS patients muscle biopsies. As for mir-133a, we found positive relationships between miR-1 levels and mRNA levels of *Myod1* in SBMA animal muscles, where *Myod1* expression was down-regulated at the late disease stage, corroborating this myomiRs/target gene pair implication in the altered molecular mechanisms underlying muscle impairment. Together with miR-206, miR-133a, and miR-133b, and differently than in G93A-SOD1 and AR113Q mice, miR-1 expression was also increased in symptomatic Δ7SMA muscle, indicating a potentially specific muscle pathological signature in SMA. Interestingly, we recently reported these myomiRs’ decrease over disease course upon nusinersen treatment in serum of infantile SMA patients associated with concurrent improvement at the functional motor scale [58]. Lastly, as previously observed for miR-133a, we found an inverse correlation of miR-1 levels with *Pax7* mRNA levels, and a positive miR-1/*Mef2a* correlation.

Hence, a potential regenerative response in muscle tissue of MND animals may be triggered during disease progression, and the dysregulation of specific myomiR/target gene pairs may account for muscle impairment and inefficient repair mechanisms in MNDs. Indeed, we previously showed [16] a significant increase in hypotrophic fibers, mainly in ALS and SBMA animal model, congruent with clinical worsening [14,16]. Some of these fibers present morphological signs of degeneration, and others are smaller as possible results of a defective regenerative attempt, in line with the presence of very few regenerating muscle cells [16,74]. Although some hypotrophic fibers were detected also in SMA skeletal muscle, general fiber morphology was mostly preserved in Δ7SMA mice. Here, as mentioned, a defective activity of the satellite cells may be responsible for the failure to mature to muscle fibers [35], in line with dysregulated myomiR/target gene patterns that we detected.

Since miRNAs are stable in body fluids and may reflect distinct pathophysiological states, they represent promising biomarkers for MNDs. These molecules can be released into the circulation by pathological affected tissues and display remarkable stability in body fluids. Indeed, in a previous study it was demonstrated that miR-206 was up-regulated in skeletal muscle and plasma of SOD1-G93A mice and in serum of ALS patients, suggesting that this molecule can reflect the pathological state of muscle in the body fluids [59]. This phenomenon was observed also in SMA mice and patients, where an up-regulation of miR-206 was detected in skeletal muscle tissue and serum [73]. Biomarkers need to be less invasive possible to be feasibly detected in clinical practice. Thus, we then investigated the acknowledged myomiRs in the respective MND patients’ serum samples. We found increased levels of miR-206 in ALS, as previously reported [53,55], and notably, also in SMA sera, along with a similar trend in SBMA patients. Thus, similar molecular regulatory mechanisms are triggered in response to pathological processes occurring in muscles, both in animal model and in humans, and miR-206 levels in serum may reflect disease status and a compensatory response to cope with muscle atrophy [75]. Our miR-206 sensitivity and specificity results supported its possible usefulness for monitoring disease progression, and it deserves future investigations. In this view, a serum decreased expression of myomiRs after therapeutic treatment might be the result of an improvement of muscle conditions and performance [58]. Likewise, in the field of ALS, Pegoraro and colleagues recently found a significant decrease in serum levels of miR-206 after physical training in ALS patients associated to stabilization of skeletal muscle and neuromuscular junction [76].

Our overall data highlight the potential of myomiRs as biomarkers and possible therapeutic targets in ALS, SMA, and SBMA. Greater understanding in this research area could have relevant implications for MND management.

## 4. Materials and Methods

### 4.1. Animal Models

All animal experiments were carried out in accordance with the EU Directive 2010/63 and with the Italian law (D.L. 26/2014) on the protection of animals used for scientific purposes. Transgenic G93A-SOD1 (B6SJL-Tg (SOD1*G93A)1Gur/J) [MGI: 2183719] and control B6.SJL mice were purchased from Charles River Laboratories, Inc. (Wilmington, MA, USA), maintained and bred at the animal house of the Fondazione IRCCS Istituto Neurologico Carlo Besta in compliance with institutional guidelines. The project was approved by the Ethics Committee of the Institute and the Italian Ministry of Health (ref. IMP-01-12; 183/2018-PR: date 03/2018-10/2021). Transgenic G93A-SOD1 progenies were identified by quantitative real-time PCR amplification of the mutant human *SOD1* gene as previously described [16]. G93A-SOD1 male animals carrying more than 27 mutant *SOD1* copies were included in the study. They were sacrificed for tissue collection by exposure to CO_2_ at week 7–8–10 (presymptomatic stages of disease), week 12 (onset of disease), week 15 (symptomatic phase of disease), and week 18 (late stage of disease) [77].

The Δ7SMA transgenic mice were adopted as in vivo model of SMA (stock no. 005025; Jackson Laboratory, Bar Harbor, Maine, USA) [MGI: 109257]; heterozygous mice (*Smn*+/−, h*SMN2*+/+, *SMNΔ7*+/+) were bred and pups were identified by genotyping, as previously reported [73]. *SMNΔ7*+/+ male mice were euthanized for tissue collection at day 2 (presymptomatic stage of disease) and day 10 (symptomatic stage of disease). Two time points were identified because the lifespan of Δ7SMA mouse model is very short (13.3–15 days) and it is very difficult to identify the onset of the disease [78]. Age-matched unaffected littermates were used as controls. All animal experiments received approval by the Italian Ministry of Health review board (1007/2016-PR and 96/2016-PR: date 2016-2019).

KI-SBMA model [MGI: 88064] was developed by Lieberman laboratory [39]. In these mice, a portion of the coding region of mouse *Ar* exon 1 (Pro37 to Gly423) was exchanged for the same region in human *AR* exon 1, introducing 113 CAG repeats. Hemizygous female mice carrying AR113Q in the X chromosome were crossbred with C57Bl/6J male mice to maintain the colony. Generation and genotyping of AR113Q knock-in mice was conducted as previously described [14,39]. Animal care and experimental procedures were conducted in accordance with the institutional guidelines (Università degli Studi di Milano), and the project was approved by the Italian Ministry of Health (423/2015-PR: date 2015-2018). The male animals were sacrificed for tissue collection by exposure to CO_2_ at weeks 8 (presymptomatic stage of disease), 14 (onset of disease), and 40 (symptomatic phase of disease) [39]. Disease stage was confirmed by monitoring body weight, rotarod performance test, and the grip strength of the animals.

Denervated animal model and control are described in Appendix A.

### 4.2. Quantitative Real-Time PCR to Assess MyomiR Expression in Mouse Muscle Tissues

At sacrifice, gastrocnemius muscles were collected, snap frozen, and maintained at –80 °C until use. Total RNA was extracted with Trizol reagent from gastrocnemius muscle tissues (100–200 mg) and reverse-transcribed to cDNA using TaqMan microRNA Reverse Transcription Kit (Thermo Fisher Scientific Inc., Foster City, MA, USA) with specific primers for miR-206, miR-133a, miR-133b, and miR-1. cDNA aliquots corresponding to 15 ng total RNA were amplified by quantitative real-time PCR in duplicate, with Universal PCR Master Mix and specific predesigned TaqMan MicroRNA assays (Thermo Fisher Scientific Inc.). U6, which was stably expressed across the G93A-SOD1, Δ7SMA, AR113Q, and control muscle tissues (as shown by standard deviation of Ct values < 0.5; Ct range: from 22.91 to 23.07), was used as endogenous control for data normalization. The range of Ct values for the miRNAs across the MND samples were the following: (i) miR-206, from 18.32 to 24.43; (ii) miR-133a, from 16.65 to 21.34; (iii) miR-133b, from 16.56 to 28.77; (iv) miR-1, from 17.14 to 23.56. The range of Ct values for the miRNAs across the control samples were the following: (i) miR-206, from 20.67 to 26.82; (ii) miR-133a, from 16.58 to 23.60; (iii) miR-133b, from 17.04 to 28.35; (iv) miR-1, from 17.50 to 32.20. MiRNA levels normalized to U6 were expressed using the formula 2^−ΔCt^. To represent the expression of myomiRs and their predicted targets, we first calculated the 2^−ΔCt^ by normalizing myomiR and target expression for each model and at each time point towards U6 and 18S respectively; fold change was estimated by dividing the 2^−ΔCt^ of pathological versus 2^−ΔCt^ control values of mice. Then, the log2 of fold change was shown in Figure 4.

### 4.3. MyomiR Target Gene Prediction and Analysis in Mouse Muscle Tissues

MyomiR targets were predicted *in silico* by the miRWalk and miRBase database using the default score parameters [62], and from the literature, selecting those genes muscle-associated known to be specifically expressed in skeletal muscle and to regulate the fundamental processes of myogenesis and regeneration [44,46,48,49,50,51]. For gene expression analysis, total RNA (0.5 ng), previously examined for myomiR expression, was treated for 15 min at room temperature with 1U of DNaseI (Merck, Darmstadt, Germany), and reverse-transcribed to cDNA using the HighCapacity cDNA Reverse Transcription Kit (Thermo Fischer Scientific Inc.), according to the manufacturer’s instructions. Quantitative real-time PCR was performed using the CFX 96 Real-Time System (Bio-Rad Laboratories, Segrate, Italy) in a 10 µl total volume, using the iTaq SYBR Green Supermix (Bio-Rad Laboratories), with a final concentration of primers of 500 nm, and with cDNA aliquots corresponding to 10 ng of total RNA. Primers for the selected genes, reported in Appendix A, were designed using the program Primer 3 Plus and purchased from Eurofins Genomics (Ebersberg, Germany). Data were normalized to the housekeeping gene “ribosomal protein lateral stalk subunit P0” (*Rplp0*) and expressed as fold changes using the formula 2^−ΔCt^.

### 4.4. Western Blotting Assay

To obtain total proteins, muscles (50 mg) were homogenized in lysis buffer, phosphate-buffered saline (PBS), pH 7.4, supplemented with 1% Nonidet P-40, protease inhibitor cocktail (Merck), phosphatase inhibitors (sodium vanadate 100 mM and sodium fluoride 100 mM), and EDTA (1 µM), with Tissue Lyser II and stainless steel glass beads (Qiagen, Venlo, Netherlands). Crude extracts were centrifuged for 10 min at 5000 rpm at 4 °C to remove particulate matter. Supernatant protein concentration was determined by the bicinchoninic acid method (BCA assay, EuroClone, Pero, Italy). Western immunoblot analysis was performed on 12% sodium dodecyl sulfate polyacrylamide gel electrophoresis loading 15 μg of total proteins. Samples were then electrotransferred to nitrocellulose membranes 0.45 m (Bio-Rad Laboratories) using a semidry transfer apparatus (Trans-Blot^®^ Turbo™ Transfer System, Bio-Rad Laboratories). Nitrocellulose membranes were treated overnight with a blocking solution containing 5% nonfat dry milk in TBS-T (Tris-buffered saline with 0.1% Tween 20) and then incubated with the right primary antibody overnight at 4 °C (anti-GAPDH dilution 1:10000, Immunological Science, Roma, Italy, MAB-10578; anti-PAX-7 dilution 1:1000, Thermo Fisher Scientific Inc., PA-1-117; anti-MYOG dilution 1:500, Thermo Fisher Scientific Inc., MA5-11486; anti-MEF2A dilution 1:1000, Thermo Fisher Scientific Inc., PA5-27344). Immunoreactivity was detected using secondary peroxidase-conjugated antibodies: goat anti-rabbit (Jackson ImmunoResearch, Cambridgeshire, UK, dilution 1:5000) was used to identify anti-MEF2A; goat anti-mouse (Jackson ImmunoResearch, dilution 1:5000) was used to identify all the other primary antibodies. Original representative images of western blots were reported in Appendix A. The immunoreactive regions were then visualized using the enhanced chemiluminescence detection kit reagents (Westar Antares, Cyanagen, Bologna, Italy). A ChemiDoc XRS System (Bio-Rad Laboratories) was used for the image acquisition. Optical intensity of samples assayed was detected and analyzed using the Image Lab software (Bio-Rad Laboratories).

### 4.5. Histological Analysis

Muscle tissue was obtained from the hind limb (gastrocnemius) of three G93A-SOD1 mice at week 18, three Δ7SMA mice at day 10, three AR113Q mice at week 40, and respective age-matched controls, immediately frozen in isopentane precooled in liquid nitrogen for histology and stored at −80 °C until histological analysis. For each animal, three frozen tissue sections (10 μm thick) were stained with hematoxylin-eosin (Bio Optica, Milan, Italy) and examined by optical microscopy (Nikon GMBH, Germany) at 40× magnification. Muscle fiber diameters were measured using Image Pro-Plus (Media Cybernetics, Silver Spring, MD, USA). Fiber diameter was defined as the widest transversal distance. For each hind limb muscle section examined, fiber diameters were measured in three randomly selected microscope fields.

### 4.6. Patients and Biological Samples

A cohort of 47 clinically defined MND patients was enrolled for the study, including: 14 ALS537 SOD1 mutant-patients [OMIM: #105400 Amyotrophic Lateral Sclerosis, ALS1] (8 sALS and 6 fALS), 23 pediatric SMA type II [OMIM: #253550 Spinal Muscular Atrophy, type II, SMA2] (17) and III [OMIM: #253400 Spinal Muscular Atrophy type III, SMA 3] (6), 10 SBMA patients [OMIM: #313200 Spinal and Bulbar Muscular Atrophy, X-linked 1, SMAX1] followed-up, respectively, at Neurology III Unit, Neurology IV Unit, Developmental Neurology Unit, Neurology X Unit, and genetically assessed at Unit of Medical Genetics and Neurogenetics at Fondazione IRCCS Istituto Neurologico Carlo Besta (Milan, Italy). Seventeen sex and age-matched adult healthy controls were included. Nineteen pediatric patients affected by encephalitis associated to antibodies anti-NMDA receptor, whose pathogenesis has no overlap with MNDs, were considered as other disease controls for children with SMA type II and III due to difficulty in collecting biological samples from healthy children. Patients’ clinical features are reported in Table 1. The study was performed in accordance with the ethical standards of The Code of Ethics of the World Medical Association (Declaration of Helsinki). The investigation and use of patients’ data for research purposes were approved by the Ethics Committee of Fondazione IRCCS Istituto Neurologico Carlo Besta (project identification code 92/2019: date January 2019–January 2022), in accordance with the Declaration of the World Medical Association. Written informed consent was obtained from each subject or legal representatives in case of pediatric subjects. Biological samples were stored at −80 °C in the Biobanks of Fondazione IRCCS Istituto Neurologico Carlo Besta and IRCCS Istituto di Ricerche Farmacologiche Mario Negri until use.

### 4.7. Quantitative Real-Time PCR to Assess MyomiRs in Patient Serum Samples

Total RNA was extracted with miRNeasy serum/plasma kit (Qiagen) from 250 µL of serum. The RNA was reverse-transcribed to cDNA using TaqMan MicroRNA Reverse Transcription Kit with specific primers for miR-206, miR-133a, miR-133b, miR-1, and miR-16, the latter used as endogenous control [79] since it was stably expressed in serum from patients (as shown by standard deviation of Ct values < 0.5). cDNA aliquots corresponding to 15 ng total RNA were amplified by quantitative real-time PCR in duplicate using Universal PCR master mix and specific predesigned TaqMan MicroRNA assays (Thermo Fisher Scientific Inc.). MiRNA levels were normalized to miR-16 and expressed using the formula 2^−ΔCt^. MiR-16 was stably expressed across serum samples with Ct range from 28.90 to 29.20, and standard deviation of Ct values < 0.5.

### 4.8. Statistical Analysis

The nonparametric distributed data, tested via Shapiro–Wilk test, were analyzed by Mann–Whitney test for comparison of two groups as indicated in the Figure legends. *p*-values < 0.05 were considered statistically significant. The miRWalk 2.0 and the miRBase database were employed to predict putative target genes of myomiRs. Nonparametric Spearman correlation test was applied to evaluate the correlation between expression levels of each miRNA and its target genes in muscle tissues of MND animal models during disease progression. Receiver operating characteristic (ROC) curves were used to assess sensitivity and specificity of miR-206 in serum samples as biomarker able to discriminate between MND patients and controls. GraphPad Prism version 4.0 (GraphPad Software, San Diego, CA, USA) was used for data elaboration and statistical analysis.

## 5. Conclusions

Our study provides new insights into common mechanisms involving myomiRs in pathophysiology of ALS, SMA, and SBMA. In particular, the overexpression of miR-206 represents a critical change shared by these conditions, likely related to the attempt to promote muscle regeneration processes in response to neuromuscular impairment, whereas miR-133a, miR-133b, and miR-1 do not support myogenesis under pathological conditions, as observed mainly in ALS and SBMA animal models. Thus, modulation of myomiR expression could represent a promising therapeutic approach for MNDs worthy of further investigation. The up-regulation of miR-206 in serum of ALS, SMA, and SBMA patients strengthens the role of this molecule as a noninvasive clinical biomarker, reflecting the pathophysiological state of skeletal muscle in MND-affected patients.

## Figures and Tables

**Figure 1 ijms-22-05673-f001:**
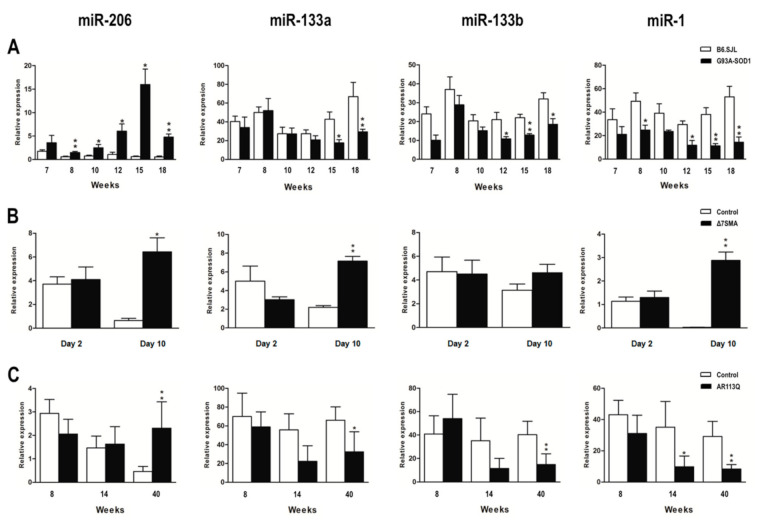
Altered expression of myomiRs in G93A-SOD1, Δ7SMA, and AR113Q mouse muscle tissue as disease progresses. Quantitative real-time PCR analysis of myomiRs in total RNA extracted from gastrocnemius muscle tissue of (**A**) G93A-SOD1 (black bars), (**B**) Δ7SMA (black bars), and (**C**) AR113Q (black bars) and control mice (white bars) at different stages of disease (5 mice per group). Relative expression data are presented as mean ± SEM of 2^−ΔCt^ values normalized against endogenous control U6. * *p* < 0.05, ** *p* < 0.01, Mann–Whitney test.

**Figure 2 ijms-22-05673-f002:**
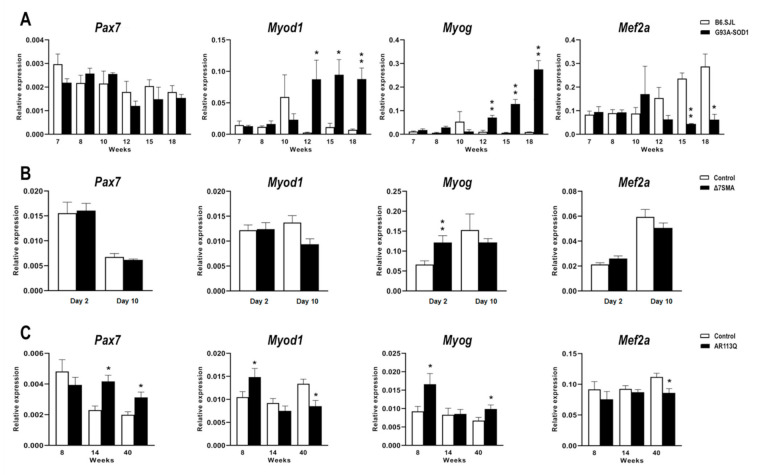
Expression levels of the predicted myomiR target genes in muscle tissue of MND and control animals obtained at different disease stage. Quantitative real-time PCR analysis of miRNA target genes, *Pax7*, *Myod1*, *Myog*, *Mef2a* in total RNA extracted from gastrocnemius muscle tissue of (**A**) G93A-SOD1 (black bars), (**B**) Δ7SMA (black bars), and (**C**) AR113Q (black bars) and control mice (white bars) (*n* = 5 mice per group). Data were normalized to *Rplp0* mRNA and expressed as 2^−ΔCt^. Relative expression data are presented as mean ± SEM. * *p* < 0.05, ** *p* < 0.01, Mann–Whitney test.

**Figure 3 ijms-22-05673-f003:**
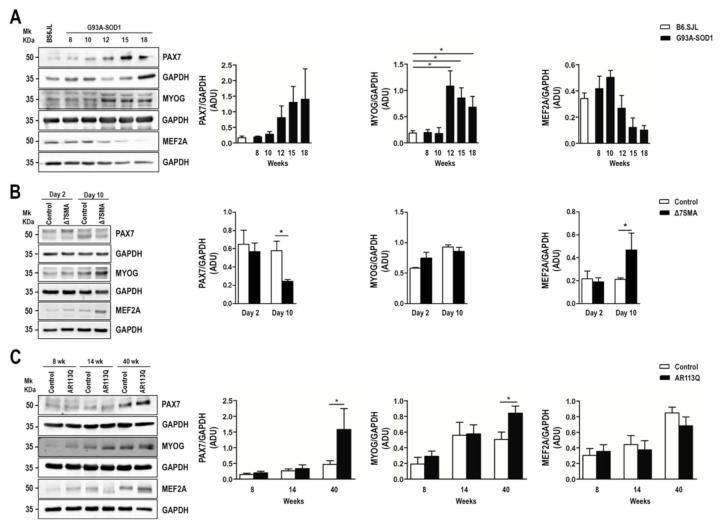
Altered expression of PAX7, MYOG, and MEF2A proteins in G93A-SOD1, Δ7SMA, and AR113Q mouse muscle as disease progresses. Representative western blot analysis of PAX7, MYOG, and MEF2A proteins in gastrocnemius muscle tissue of (**A**) G93A-SOD1 (black bars), (**B**) Δ7SMA (black bars), and (**C**) AR113Q (black bars) mice and control mice (white bars) (*n* = 3 mice per group) with relative densitometric analysis. Density values are reported as mean ± SEM, corrected for background and normalized to GAPDH control. * *p* < 0.05, Mann-Whitney test.

**Figure 4 ijms-22-05673-f004:**
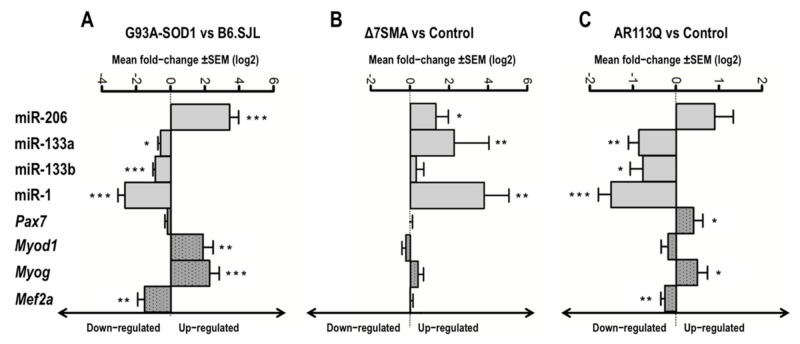
Altered expression of myomiRs and their predicted targets in G93A-SOD1, Δ7SMA, and AR113Q mouse muscle. Data are presented as mean ± SEM of log2 of fold changes of 2^−ΔCt^ expression of myomiRs (grey bars) and mRNA targets (grey with dot bars) obtained at different disease stages in (**A**) G93A-SOD1, (**B**) Δ7SMA, and (**C**) AR113Q relative to control mice. * *p* < 0.05, ** *p* < 0.01, *** *p* < 0.001, Mann–Whitney test.

**Figure 5 ijms-22-05673-f005:**
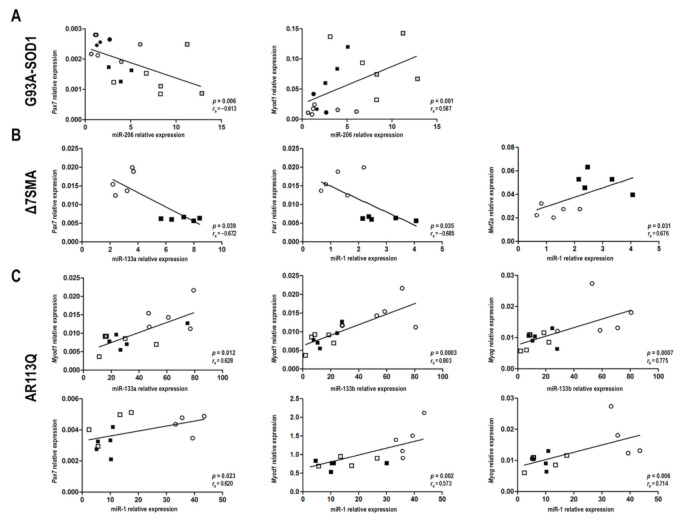
MyomiR/mRNA correlation analysis in G93A-SOD1, Δ7SMA, and AR113Q during disease progression. Negative and positive correlations (*p* < 0.05) estimated by Spearman’s correlation test between myomiR and mRNA levels of *Pax7*, *Myod1*, *Myog*, and *Mef2a* in muscle tissue of: (**A**) G93A-SOD1 at weeks 7 (●), 8 (○), 10 (●), 12 (□), 15 (■), and 18 (■); (**B**) Δ7SMA at day 2 (○) and 10 (■); (**C**) AR113Q at weeks 8 (○), 14 (□), and 40 (■).

**Figure 6 ijms-22-05673-f006:**
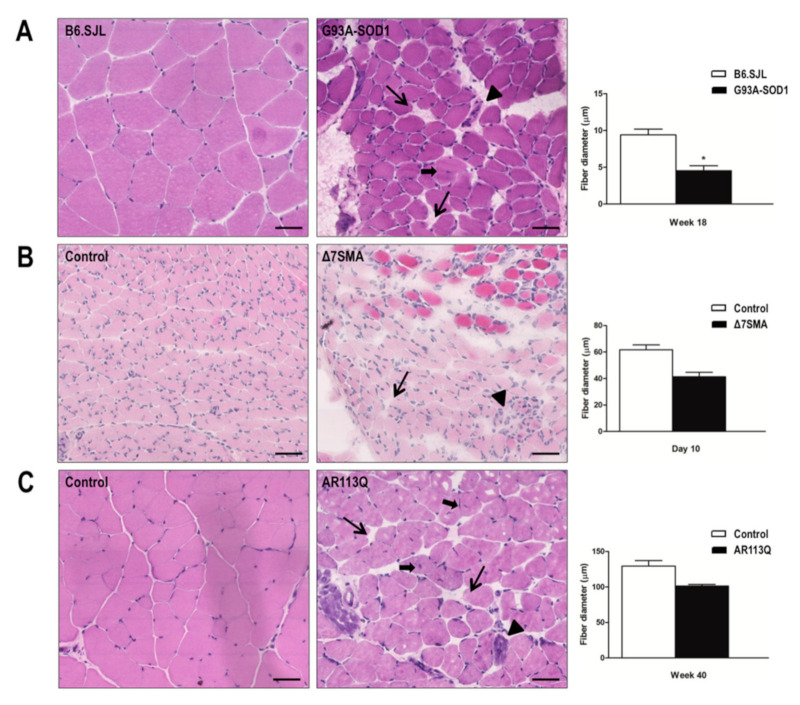
Histological analysis of gastrocnemius muscle fiber diameter in G93A-SOD1, Δ7SMA, and AR113Q mice. Representative transversal skeletal muscle sections stained with hematoxylin/eosin in (right column) (**A**) G93A-SOD1 mice, (**B**) Δ7SMA mice, and (**C**) AR113Q mice at late disease stages (week 18, day 10, and week 40, respectively), and relative age-matched controls (left column). Long arrows indicate enlarged endomysial spaces; arrowheads indicate degenerating cells with apoptotic fragmented nuclei; short arrows indicate fibers with centrally located nuclei, common during muscle regeneration. Magnification 40X. Scale bar = 50 μm. On the right, measurement of fiber diameter in gastrocnemius muscle of G93A-SOD1 at week 18 (black bars), Δ7SMA mice at day 10 (black bars), AR113Q mice at week 40 (black bars), and relative age matched control mice (white bars). Each histogram represents the mean diameter (µm) ± SEM of muscle fibers measured in three muscle sections per mouse from three mice in each group. * *p* < 0.05. Mann–Whitney test.

**Figure 7 ijms-22-05673-f007:**
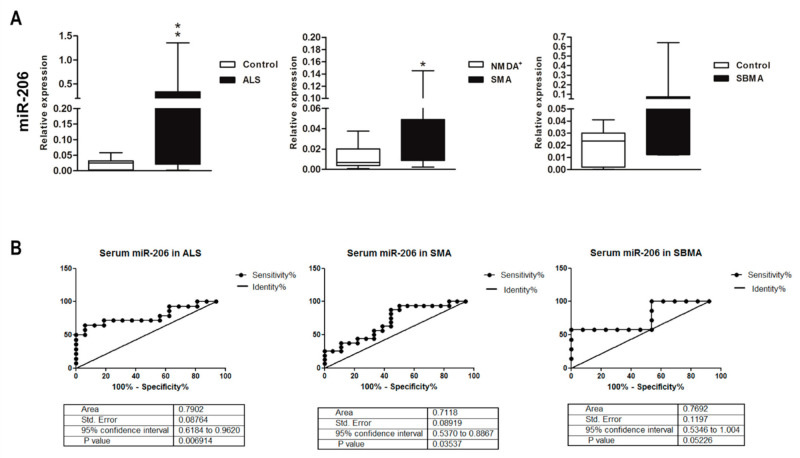
Up-regulation of miR-206 in serum of ALS, SMA, and SBMA patients. (**A**) Quantitative real-time PCR analysis of myomiRs in total RNA extracted from serum of ALS, pediatric SMA, and SBMA patients (black bars), healthy controls (white bars) and patients with anti-NMDA receptor encephalitis (white bars in the SMA graph) as controls for pediatric SMA. Relative expression data are presented as mean ± SEM of 2^−ΔCt^ values normalized against the endogenous control miR-16. * *p* < 0.05, ** *p* < 0.01, Mann–Whitney test. (**B**) Receiver operating characteristic (ROC) curves used to assess the sensitivity and specificity of miR-206 in serum as biomarker for ALS, SMA, and SBMA.

**Table 1 ijms-22-05673-t001:** Summary of the main features of ALS, SMA, and SBMA patients and controls included in the study.

	*SOD1*-Mutated ALS (*n* = 14)	SBMA (*n* = 10)	Healthy Controls (*n* = 17)	SMA (*n* = 23)	Anti-NMDA Receptor Encephalitis (*n* = 19)
Sex (F/M)	7/7	0/10	8/9	13/10	12/7
Age at serum collection (Years, mean ± SD)	50 ± 12.2	56.3 ± 5.92	48.64 ± 11.33	6.86 ± 3.33	13.13 ± 5.62
Disease duration (Years, mean ± SD)	2.73 ± 3	15.9 ± 9.48	-	5.15 ± 2.95	-
Disease-related information	Sporadic/familial SOD1-ALS 8/6	CAG expansion (38 to 62 repeats)	-	SMN1 deletion Type II/III 17/6	Autoimmune encephalitis
Muscle function at time of serum collection	14 pts with limb weakness (walkers), of which 4 with dysphagia	4 pts with fatigability; 1 pt with muscle cramps; 1 pt with limb weakness (walker); 4 pts with dysphagia and limb weakness (3 walkers, 1 sitter)	-	18 sitters; 5 walkers	-

## Data Availability

Data are available from the corresponding author on reasonable request from any qualified investigator. All patients’ data from this study will be shared anonymized in accordance with consent provided by participants on the use of confidential data. Data reuse is permitted only for academic purposes.

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
