# Peer review of "Dysregulation of Muscle-Specific MicroRNAs as Common Pathogenic Feature Associated with Muscle Atrophy in ALS, SMA and SBMA: Evidence from Animal Models and Human Patients"

_ijms, 2021, doi:10.3390/ijms22115673_

Round 1

Reviewer 1 Report

This manuscript proposes a muscle microRNA as a novel serum biomarker of disease progression in ALS, SMA and SBMA.

The authors first review data that suggest muscle wasting could contribute to neuromuscular pathologies often considered to mostly result from motor neuron degeneration in ALS, SMA and SBMA. They then present  experiments in which they investigate the expression of selected muscle-specific  microRNAs (myomiRs) during disease progression in muscles of 3 transgenic mouse models of ALS, SMA and SBMA. They further evaluate muscle expression of these myomiR putative target genes at the RNA and protein levels, and study correlations in the abundances of individual miRs and their target gene products. Finally the authors quantify the same myomiRs in the sera of patients affected with ALS, SMA or SBMA during disease progression, and conclude that miR206 serum levels could constitute a biomarker for these motoneuron degenerescence pathologies.

The manuscript is very clearly written, and the experiments well designed and performed, with appropriate sample numbers, controls and statistical analyses. The discussion is excellent and takes into account a large number of publications in the field.

Major comments

  1. For my major comment I shall focus on miR206 because of its proposed role as a disease biomarker. The authors conclude from their first experiments that the observed miR206 induction in skeletal muscles of the transgenic mouse models is associated to disease progression and reflects attempts at muscle regeneration. The next point in their study is to directly investigate miR206 levels in patient sera. However a step is missing in their reasoning: what is the evidence that myomiR serum levels do reflect their skeletal muscle levels in each of these mouse models during disease progression? The studied myomiRs should be quantified in the sera of the 3 mouse models and putative correlations with muscle myomiR levels and disease progression investigated. Although good correlations will probably be obtained, the experiment has to be done since the impact of motoneuron degeneration and muscle alterations (pathological changes and attempts at regeneration) on myomiR release in blood is unknown.
  2. MiRs have different modes of actions on their target messenger RNAs: they either cause mRNA destruction via the RISC complex, inhibit or activate mRNA translation. Could the authors comment on that point in their data interpretation in the discussion?

Minor comments:

1. Gene/protein nomenclature should be checked all over text, tables and Figures, including supplemental data:

Pax7 is the mouse gene, while PAX7 is the human one (e.g. FigS2)

Pax7 is the mouse protein, while PAX7 is the human one (e.g. line 229)

2. what is the meaning of RT in quantitative RT-PCR? Is it real time or reverse transcription? What one is assaying is the amount of mRNA, so a reverse transcription should be mentioned even if the quantification is done by real time during PCR amplification of the resulting cDNA.

lines 129-131: these micro RNAs act on messenger RNAs, not on genes

line 133 and in discussion: a reference is missing about earlier observation of miR206 increase in ALS muscles (Pegoraro et al 2020 Clin. Neuropathol. 39, 105)

line 163: mouse muscle tissue

lines 180 , 183:  correct English: "both the two"  => "both" 

line 191: English: disclosed => discovered, detected, observed

lines 227-228: confusion between genes and proteins (see point 1 above)

Figure 4: could the authors give an example (in the Method section and in FigS2) of the calculation they used to take into account the RT-qPCR data at all time points for myomiR and their target expression?

Line 288: English: Derangement => alteration, disruption?

Line 290: in the evaluation of muscle pathology couldn't the authors perform immunostaining for developmental myosin (Schiaffino et al 2015;  Skel.Muscle 5, 22) as an evidence for on-going regeneration instead of relying on fibers with central nuclei?

Line 479: English: a CAG repeat of 113 => 113 CAG repeats or (CAG)113

Line 503: English (latin!): predicted in silico

Lines 504-505: please clarify the sentence "strictly implicated...skeletal muscle"

Figure S2: in order to easily compare the data presented in the result section in the 3 transgenic mouse models and those in the denervation model here, could the quantifications be presented at the different times instead as an average of values at both times.

Denervation mouse model in Supplementary Material: could the authors clarify what control muscle was used? In the method part it is stated  "Right sciatic nerve was only exposed and utilized as sham internal control in each animal" while in FigS2 "control animals" are mentioned.

Reviewer 2 Report

Malacarne et al., hypothesized that miRNAs that are differentially in muscles from mouse models for three NMDs could mark disease progression (progressive muscle degeneration). They show expression levels of four myomiRs in 2, 3 or 6 time points representing “pre-symptomatic, onset and the symptomatic phase”, using qRT-PCR. The “symptomatic phase“ was complemented with H&E staining of the muscle. They show results from the gastrocnemius muscle. Using miRWalk2.0, they then selected 4 predicted miR-targets, assessing RNA levels and protein levels from the same muscles and compared them to the selected myoMiRs levels. Last, they show expression levels of miR-206 in serum from patients, as a biomarker for these MNDs. They suggest that those myomiRs can be clinical biomarkers for these MNDs and that these myomiRs could be therapeutic targets for MND treatment.

General comments:

The hypothesis: searching for common miRNAs for NMDs is interesting and relevant. Focusing on muscle tissue as a commonly affected tissue is reasoned well, yet, the choice for the gastrocnemius muscle is not indicated, and progression of muscle atrophy or muscle degeneration is not shown. The histological description of the affected muscle is quite poor: muscle degeneration includes several whole marks: inflammation, central nuclei (which marks degeneration), ECM thickening, fatty infiltration, and a switch in myofiber type, as well as a change in the myofiber CSA. The myofiber CSA shows a large range of values, and the range should be shown. In general, the contrast seen with H&E staining is not sufficient for accurate segmentation of myofibers. The authors show only one point of tissue histology, without a progression in muscle degeneration. Therefore, it is unclear how the authors defined the three phases: “pre-symptomatic, onset and the symptomatic phase”.

A minor comment to the general statement of three phases: only two-time points are shown for the Δ7SMA model, thus not three phases.

The miRs were assessed with qRT-PCR and levels were calculated after normalization to a single gene. qRT-PCR of miRNAs is highly affected by technical issues, and therefore most studies add spike (preferably two) and if normalized to a ‘housekeeping’ miRNAs, then two are included, and the consistency across samples should be demonstrated by the CT values. In addition, the range CT values for the miRNAs should be reported. The ddCT normalizes first to the loading control and then to the biological control, thus reporting the fold change. It is not clear how the ddCT was calculated, as the control bar (time 0) should be at 1.

The choice for the 4 putative targeted genes is not convincing. I suggest looking at this paper https://journals.sagepub.com/doi/full/10.4137/BMI.S29513. Moreover, the authors selected 4 genes regulators of myogenesis and regeneration, but they do not show regeneration in the affected muscles. On the other hand, regulators of muscle atrophy were not selected. This seems like an arbitrary selection. In most cases, changes in RNA levels are not in agreement with changes in protein levels. Only one set of the Western blots is shown, the data in the supplementary is the same as in Figure 3. Minor: in one blot GAPDH is noted as 35 KD and in one blot 40 KD.

Figure 4 already suggests that there is little correlation between the miRs and the target genes. In brief: higher miRs levels should result in lower expression of the targeted gene (if it is directly regulated). Considering that, in Figure 5, only the dot-plots for Pax-7 can be considered as relevant. Yet, Pax-7 expression levels do not change in both G93A-SOD1  and d7SMA muscles, suggesting that the correlation is age-associated but not associated with these genotypes. Minor: the statistical test must be annotated in the Figure legends.

From the 4 myomIRs only miR-206 remained significant in serum from patients. This is not surprising, as miR-206 has been reported in many muscular disorders. Therefore, it cannot be a specific biomarker for MNDs. That PCR study lacks the same controls as mentioned above. The information “age of onset” is not sufficient, the age of serum collection is important, and information on muscle function should be added. Many ALS patients come to the clinic before they are wheel-chaired.    

Minor: The table presentation should be improved: samples with the same control group should be presented together, otherwise confusing.

The introduction and discussion are both interesting but have little relevance to the results (both are read as if it is a review).

Reviewer 3 Report

Malacarne and colleagues present a convincing set of data to support the conclusion that muscle-specific microRNAs are dysregulated in motor neuron diseases. Working with mouse models of ALS, SMA and SBMA, they highlight some consistent and some discordant changes in the levels of microRNAs and their targets in diseased muscle. They also show similarities with results generated from experimentally denervated mice, indicating that the observed changes are likely a secondary consequence of neurodegeneration. Finally, the authors identify miR-206 as a potential serum biomarker in patients across motor neuron diseases. Overall, the work is well presented, with clear figures and narrative, and will be of use to the field. However, I have several comments that should be considered before the manuscript is published:

  • To aid comprehension, the authors should consider exchanging ‘myomiRs’ in the title with ‘myo-miRs’, ‘muscle-specific microRNAs’, ‘myogenic miRNAs’, or an equivalent.
  • The Introduction and Discussion would benefit from having several paragraphs as opposed to single, rather long chunks of text.
  • Mouse, rather than human, nomenclature is used when first mentioning genes in the Introduction and should be corrected (e.g., Smn1 is used rather than SMN1, Sod1 instead of SOD1, etc.)
  • Lines 80-81: Mutations in SMN1 can indeed result in production of truncated SMN protein – but deletion mutations can also result in a complete lack of SMN protein.
  • Lines 92-94: It was mentioned that there is a common involvement of skeletal muscle in MNDs, which is subsequently discussed and well referenced. However, for completeness, I think it would be useful to also briefly mention the evidence for mechanistic similarities/overlap between these conditions. Several recent reviews have done so for SMA and ALS: https://pubmed.ncbi.nlm.nih.gov/29313812/, https://pubmed.ncbi.nlm.nih.gov/29270111/
  • Re-arranging Figure 1 to present the disease models in rows rather than columns, like the subsequent figures, will aid interpretation and consistency across the manuscript.
  • Figure 1-3: I think there is a good argument for re-analysing the data in Figures 1-3 using a two-way ANOVA or non-parametric equivalent, as opposed to the multitude of individual Mann Whitney U-tests.
  • Providing experimental details of the denervation approach when first introducing the method (lines 190-192) will increase understanding of the reason for its inclusion.
  • The key for Sod1 mice in the legend of Figure 5 does not work (e.g., 7 and 10 weeks are both black circles, whereas 15 and 18 weeks are both black squares).
  • Figure 6: rather than arbitrary units, having diameters in µm would be more informative and allow comparisons across models.
  • Lines 325-327: could the authors please provide these data as a supplementary figure so that readers can get a better understanding of variability?
  • Please provide antibody catalogue numbers.
  • The Kruskall-Wallis test with Dunn’s post-hoc test is a rather strange choice for the data in the right hand panel of Figure S2 – would a two-sample test not be more appropriate? Also, SEM is included on all graphs except for the Western densitometry in this figure – can the authors clarify the reason please?
  • Were the data in Figure S4 statistically analysed? Please do so and provide details of the test.
  • Discussion of the n = 3 Western blot data (Fig. S2 and 3) being potentially underpowered should be included.

Round 2

Reviewer 1 Report

This submission by Malacarne et al is the revised version of a manuscript I have previously reviewed in which the authors investigate myo-miR and their target gene expression in muscles of ALS, SMA and SBMA mouse models during disease progression with the hypothesis that muscle is affected early in these motor neuron pathologies. They further study these myo-miRs in the sera of patients and suggest they could constitute new biomarkers of disease progression, with a special focus on increased miR206 levels.

The authors have answered all my criticisms, and I think those of the other reviewers as well. There are just small English mistakes in the new text, 

lines 90-91: "it have been reported" => "it has been reported"

line 428: "resulted up-regulated" ??

line 511:  suppress "a" in "introducing a 113 CAG repeats"

line  542 and in the supplemental data : "fold change was showed" instead of "was shown"
